# Sources of work-related psychological distress experienced by UK-wide foundation and junior doctors: a qualitative study

Ruth Riley ![ORCID],[1] Marta Buszewicz,[2] Farina Kokab,[3] Kevin Teoh ![ORCID],[4] Anya Gopfert,[5] Anna K Taylor,[6] Maria Van Hove,[7] James Martin,[8] Louis Appleby,[9] Carolyn Chew-Graham[10]

► http://dx.doi.org/10.1136/bmjopen-2020-045588

For numbered affiliations see end of article.

**Correspondence to**
Dr Ruth Riley;
r.riley@bham.ac.uk

## ABSTRACT

**Objectives** This paper reports findings exploring work cultures, contexts and conditions associated with psychological distress in foundation and junior doctors.

**Design** Qualitative study using in-depth interviews with 21 junior doctor participants. The interviews were audio-recorded, transcribed, anonymised and imported into NVivo V.11 to facilitate data management. Data were analysed using a thematic analysis employing the constant comparative method.

**Setting** NHS in England.

**Participants** A purposive sample of 16 female and five male junior doctor junior doctor participants who self-identified as having stress, distress, anxiety, depression and suicidal thoughts, or having attempted to kill themselves.

**Results** Analysis reported four key themes: (1) workload and working conditions; (2) toxic work cultures—including abuse and bullying, sexism and racism, culture of blaming and shaming; (3) lack of support; (4) stigma and a perceived need to appear invulnerable.

**Conclusion** This study highlights the need for future solutions and interventions targeted at improving work cultures and conditions. There needs to be greater recognition of the components and cumulative effects of potentially toxic workplaces and stressors intrinsic to the work of junior doctors, such as the stress of managing high workloads and lack of access to clinical and emotional support. A cultural shift is needed within medicine to more supportive and compassionate leadership and work environments, and a zero-tolerance approach to bullying, harassment and discrimination.

## Strengths and limitations of this study

► Few qualitative studies have explored work-related psychological distress in junior doctors; therefore, we employed in-depth interviews to explore such distress, interviewing 21 junior doctors from across the UK.

► The sample included five male participants compared with 16 female participants, which may be a limitation in this study.

► Participants were self-selecting, which may be perceived as a limitation; however, the purposive sampling of participants ensured that the sample was varied in terms of ethnicity, number of years in training, specialty and geographical location.

► Semistructured interviews generated rich qualitative data, and an iterative process of concurrent data collection and constant comparative analysis facilitated the simultaneous exploration, refinement and enrichment of key themes.

► The analysis was conducted by a multidisciplinary team comprising three junior doctors, two academic clinicians, an occupational health psychologist and three social scientists which afforded checks and balances to the validity of the analytical process, interpretation of data and transferability of the research findings.

## INTRODUCTION

There are currently 115 376 doctors working in the NHS, almost half of whom (56 404) are 'junior doctors', including doctors in training or at preconsultant grade, and foundation doctors.[1] Previous research indicates that doctors worldwide are vulnerable to chronic stress, anxiety, depression and burnout.[2–4] High rates have frequently been reported but with considerable heterogeneity over time and between countries.[5] Despite mixed findings about suicide rates internationally, there are concerns over suicidal ideation and behaviour in doctors, particularly among female doctors.[6 7]

The Practitioner Health Programme in London, an England-wide specialist mental health, drug and alcohol service for doctors and dentists, reports that two-thirds of those presenting are women, with higher attendance rates among young female doctors.[8] It is uncertain whether this is representative of the general population of doctors or reflects gender differences in help-seeking

behaviour. A review and meta-analysis of international research found that being female has been identified as a risk factor for stress and burnout among junior doctors, but explanations for this trend remain inconclusive.[9]

Most previous research into distress and suicide in junior doctors uses quantitative, retrospective studies, with the focus on the individual (particularly emphasising the need for 'resilience'),[10 11] and overlooking the wider systemic, organisational and cultural sources of distress among this population of health professionals. This approach is likely to detract from considering solutions/interventions which target organisational culture and systemic factors.[12] To our knowledge, no previous study has qualitatively examined how junior doctors view their working conditions and work cultures, and their relationship to elevated psychological distress.

Factors contributing to distress and mental health conditions such as depression, anxiety and suicidality, or which compound stress or underlying mental ill health among doctors, have been identified through recent research exploring mental illness and help seeking among general practitioners (GPs).[13] Participants attributed sources of stress/distress to systemic factors relating to increasing workloads, bureaucracy, loss of autonomy and toxic work cultures, including dysfunctional relationships at work, bullying, lack of support and isolation.

A recent systematic review found that work-related factors such as work demands, worries about patient care, poor work environment and poor work–life balance were associated with higher rates of stress/burnout among junior doctors.[9] Universally, junior doctors face specific pressures related to their professional stage and development. In a nationwide survey of junior doctors in Ireland, stress, anxiety and depression were highest among junior doctors as compared with consultants.[14] Such pressures are compounded by working in healthcare systems which face escalating demands, diminishing resources and staff shortages.[15]

A study exploring junior doctors' reasons for leaving medicine in the UK cited lack of support, mentorship or formal training, loneliness and bullying.[16] Feeling undervalued, unsupported or having reduced autonomy were key factors affecting low morale among junior doctors[17]; this evidence highlights the impact of working conditions and work culture on junior doctors and the potential value of providing effective support and supervision.

Identifying such patterns or contextual factors is crucial in developing targeted improvements or interventions that can reduce the risk of mental ill health and psychological distress from the outset. This paper reports findings as part of a wider mixed-methods study exploring work, cultures, contexts and conditions associated with psychological distress in this population. This qualitative study aimed to identify and describe work-related causes/sources of psychological stress/distress experienced by junior doctors.

## METHODS
### Study design and setting
We employed an interpretative methodology using qualitative methods, with semistructured interviews to explore junior doctors' perspectives and experiences of stress and distress. The study setting was NHS England.

### Sampling and recruitment
We used a range of recruitment methods, using advertisements in social media (eg, the British Medical Association (BMA) junior doctors' Facebook group comprising 56 698 members, through Twitter and the mental health research charity websites). In addition, information about the study was circulated through the Practitioner Health Programme, a specialised mental health and drug and alcohol service dedicated to doctors and dentists.[18]

Potential participants were asked to express their interest by contacting the study team who then provided further information about the study.

The research team employed purposive sampling to ensure maximum variation, taking account of the following characteristics: gender, age, ethnicity, sexuality, geographical location, different grades/duration of General Medical Council (GMC) registration, medical specialty, disclosure of a mental health diagnosis, individuals who reported self-harm behaviour, had thoughts of suicide or had attempted to kill themselves.

The eligibility criteria used are listed in table 1.

When a junior doctor agreed to participate, they were sent a reply slip and consent form electronically and asked to sign the consent form prior to the interview, before returning it to the study researcher. Face-to-face, telephone or 'Skype' interviews were arranged at a time convenient to each participant. They were given the opportunity to raise questions prior to being interviewed.

At the end of each interview, the researcher informed the participant about the next steps and how their data would be used, and checked on their well-being prior to leaving the interview setting. A risk protocol was used to ensure appropriate support was provided to participants in the event of the disclosure of suicidal ideation.

### Data collection
A topic guide (see Box 1) was developed by the research team to generate discussion in the semistructured interview and modified iteratively as data collection and analysis progressed. The topic guide aimed to capture participants' views, experiences, feelings and beliefs about working conditions and cultures which were perceived to be stressful or distressing. We also aimed to identify potential protective factors (eg, existing interventions such as regular debriefs, policies and available support). The topic guide was informed by the existing literature, input from junior doctors on the study team and patient and public involvement consultation exercises conducted prior to obtaining funding.

One-to-one interviews were conducted face-to-face, by telephone or by Skype and recorded digitally with consent.

**Table 1** Inclusion and exclusion criteria

| Inclusion criteria | Exclusion criteria |
|---|---|
| Currently a foundation or junior (preconsultant) doctor working in a hospital in the NHS in England/Wales within the last 2 years—this covers any preconsultant doctor working in the NHS. They do not necessarily have to be on a training contract. | Currently experiencing acute severe mental illness such as psychosis |
| Experience (within the last 4 years) of stress, distress, mental illness, self-harm (eg, cutting and overdoses), suicidal thoughts, feelings and intent | Currently in receipt of drug and alcohol services (this does not include doctors who currently use alcohol/drugs to self-medicate, such as doctors who may be using alcohol/drugs on a regular basis to help them relax but would not be classified as having an addiction problem) |
| Has capacity to provide informed consent (this will be assumed given they are healthcare staff) | Actively suicidal or anyone who has made a suicide attempt within the last 6 weeks—to avoid causing any additional psychological harm during a period of acute distress/vulnerability |

The in-depth interviews were conducted by two authors (FK and RR), both female social and behavioural scientists with qualitative methods expertise. The recorded interviews were transcribed verbatim and checked for accuracy of transcription by the study researcher before analysis. All transcripts were anonymised before discussion within the wider research team. Reflexive notes were recorded by researchers throughout the process. Recruitment and data collection were continued until data saturation was judged to have been achieved.[19]

### Data analysis
Analysis began with data collection and was iterative, employing the constant comparative method[20 21] until data saturation was achieved, such that no new analytical categories emerged.[19]

The coinvestigators included three junior doctors (AG, AKT and MVH), two academic GPs (CC-G and MB), one occupational health psychologist (KT) and one social scientist. FK coded the data set; RR coded a subsample and contributed to the organisation of themes. The multidisciplinary provided commentary on transcripts to generate and refine codes and thematic categories and provide researcher triangulation, thereby increasing the credibility of the research findings.[22] Data were managed using NVivo. A stakeholder advisory group was involved throughout the research process. The study is reported in line with the Consolidated Criteria for Reporting Qualitative Research.[23]

---

**Box 1    Interview topic guide**

**Introduction and background**
► Describe general less stressful/more stressful jobs and the difference between these.

**Work environment**
► Describe main sources of stress in day-to-day working life.
► Explore wider sources or stress not already mentioned.
► Explore which jobs are more stressful.

**Impact of work on mental health and well-being**
► Past, present and future outlook as to how work impacted/may impact mental health and well-being.

**Preventing/seeking/managing help**
► Explore management of workload/stress in day-to-day work life.
► Discuss help seeking for distressing events.
► Explore relationships with colleagues and how concerns are responded to if raised.

**Experience of help seeking**
► Explore thoughts/feelings of seeking help.
► Explore knowledge of available support.

**What could make things better?**
► Explore any realistic changes at individual and organisational levels and upstream changes.

---

### Patient and public involvement and engagement (PPIE)
There are three junior doctors on the research team, all of whom consulted with colleagues about the initial research idea. Five junior doctors gave feedback on the initial funding application, and four junior doctors gave feedback on the protocol, topic guide and participant facing documents. We are working on a dissemination strategy with junior doctors outside the research team. Due to the time constraints of junior doctors, PPIE members were consulted via email and telephone.

### Reflexivity
RR is epistemologically steeped in qualitative traditions underpinned by interpretivism and phenomenology and oriented by critical theory such as feminism. With a background in psychology and sociology, and as a non-clinician, RR's interest in work cultures and conditions may also have been influenced by her experience of working as a researcher and medical educator where she has observed and experienced rationalist and hegemonic cultures in which there is an intolerance of vulnerability. Such experiential and epistemological orientations are likely to have influenced this topic and an interest in exploring why female doctors are more likely to experience distress.

    

**Table 2**  Participant characteristics

| Participant characteristics | N=21 |
|---|---|
| Sex (female) | 16 |
| Age (years) | |
| 20–29 | 10 |
| 30–39 | 11 |
| Ethnicity | |
| Asian (other) | 2 |
| Bangladeshi | 1 |
| Chinese | 1 |
| Indian | 3 |
| White | 13 |
| White (other) | 1 |
| Sexual orientation (heterosexual) | 15 |
| Years since qualification | |
| 0–5 | 10 |
| 6–10 | 9 |
| 11–15 | 2 |
| Specialty | |
| Emergency medicine | 2 |
| Medicine (including acute, diabetes/endocrinology and geriatrics) | 9 |
| Obstetrics and gynaecology | 6 |
| Paediatrics | 2 |
| Psychiatry | 1 |
| Public health | 1 |

FK has a constructivist and interpretivist epistemological perspective that helps her understand meanings applied to phenomena by individuals. Similarly, with a background in psychology and social research and working as a non-clinical researcher in a medical educational setting, FK understands clinical settings and as part of her doctoral research explored hegemonic masculinity in the context of accessing healthcare by men in migrant community groups.

## FINDINGS

Twenty-one interviews were conducted with participants, lasting between 43 and 103 min (mean=65 min), between November 2019 and May 2020. The demographic and professional characteristics of participants are included in table 2. Analysis of the interview transcripts and field notes identified four main themes, with corresponding subthemes relating to sources of work-related stress/distress:

► Workload and working conditions.
► Toxic work cultures—abuse and bullying, sexism and racism, culture of blaming and shaming.
► Lack of support.
► Stigma and a perceived need to appear invulnerable.

## Workload and working conditions
### Workload
Participants reported that excessive workloads were a key source of distress and emotional exhaustion, such as managing unrealistic workloads and long hours, compounded by understaffing and the frequent inaccessibility or unapproachability of senior staff to provide clinical guidance and support. Participants, particularly foundation doctors, frequently reported role conflict, with an incompatibility between the demands and the individual's level of experience or position within the organisation, which was a clear source of work stress:

> I think as juniors, we felt like no one had ever taught us how to be a doctor [right, yeah], and the pressures for things like on-calls and looking after hundreds and hundreds of patients when you're new to it, is just quite phenomenal really. There were times when one of the nurses would find me crying in a corner, you know, just upset because I didn't know what to do on a nightshift or something, I felt unsupported. (JD09, female)

### Working conditions
Participants highlighted considerable variability in the provision of appropriate spaces or basic facilities (eg, on-call rooms, doctor's mess) to talk/de-brief/off-load/rest, which contributed to feelings of exhaustion, demoralisation and stress. Coupled with poor pay these were key motivators of an intention to quit medicine for some participants.

> So many colleagues have left the profession over the past 10 years because of the working conditions and pay, you know massive pay cuts really in real terms. And just the conditions are so poor, terrible buildings, no office space, nowhere even to put your stuff, hang your coat, you know like complete basic, lack of basic facilities. (JD 23, male)

Many participants reflected that on-call rotas were a physically and emotionally demanding aspect of the job. Exhaustion related to the pressures and long hours, with some participants working 12-hour shifts, was felt to be a key contributor to stress, as this participant highlights:

> And the on-call rota, like it's physically very demanding. You're on-call for you know, 12 hours a day for like 4 days in a row… I think that exhaustion is probably what's making me feel so much more stressed than in medical school. Because in medical school I was never exhausted. I was just under pressure. Whereas now I'm under pressure and am tired all the time. (JD14, Female)

## Toxic work cultures
The most frequently reported source or cause of stress/distress experienced by study participants was attributable to toxic or unsupportive work cultures related to, namely,

bullying, sexism and discrimination, a blame and shame culture and fear of whistleblowing.

## Abuse and bullying

Over half the study participants, both men and women, reported experience of abuse and being bullied. This included behaviours, often perpetrated by consultants, but also occasionally by nursing and midwifery staff, which were perceived by participants to be devaluing, undermining, patronising, intimidating and humiliating, and were often conducted in front of other colleagues and sometimes patients. For some participants, the bullying or toxic work cultures often contributed to, or, compounded existing work-related stress/distress and/or was the tipping point or trigger for anxiety, depression and suicidal thoughts.

The following junior doctor experienced systematic bullying during one rotation, which was the source of significant distress and was the precursor to depression and suicidal thoughts:

…so, use of language gets very unprofessional so I was called stupid, I was made some horrible remark such as, 'How on earth did you pass finals? How on earth did you get into med school?', things like that. … so asking personal questions in a workspace environment repeatedly…another examples are I mean one thing is explaining things to me that I already know but in a patronising voice and then criticising me almost for their perception of me not knowing it even though I do know it [and] using very aggressive language. They raise their voice or they may even humiliate you in a public setting like in a handover meeting. (JD10, male)

Similarly, the following participant recollected her experience of being on a demanding rotation, with a heavy workload and responsibility for 15–20 patients, working long hours and feeling under pressure, with little opportunity to eat, rehydrate or rest. She recalled that her seniors and consultant were overtly critical and taught by humiliating other junior doctors. This experience subsequently triggered an episode of depression and suicidal thoughts:

Some of the seniors and some of the consultants that I had to work with gave you a really hard time as well and made you feel completely incompetent. It broke me. In fairness, it wasn't just me. When I talk to quite a few of my colleagues, they say it broke them too during that time …. I just became really low, really depressed and suicidal again. (JD22, female)

Some participants reported a cultural tolerance and acceptance of bullying in the workplace, where junior doctors were expected to put up and shut up, including being expected to do this by their supervisors:

…there was a lot of drama within the department, so, a lot of kind of bitching and bullying and even

though I did raise some issues with my supervisors, he was a pleasant fine man, that wasn't involved in any of the bullying, it was very much felt like, well, this is how the department is, suck it up and deal with it. (JD20, female)

One junior doctor participant reflected on the personal and professional cost of bullying; she reported that the bullying escalated because of her whistleblowing and, after making her complaint, she was moved early to her next rotation. This participant reported that this resulted in senior colleagues blemishing her career portfolio by providing negative feedback:

I had good feedback in my following placements, but you know I've got a blemish on my portfolio there that I can't remove and it's a direct result of bullying [yeah] and their inability to take feedback and just yeah just put the blame on me because I've spoken up so that's the main bad experience I've had so far. (JD18, female)

## Sexism and racism

Some female participants had experienced sexism in the workplace. These were often micro aggressions, such as subtle undermining comments, inappropriate sexist comments masquerading as jokes, being ignored, invalidated or not taken seriously:

…there are certainly a number of -mainly older- male consultants who are just, they have not moved with the times, they are still a bit inappropriate towards female colleagues and either just sort of either not controlling where their eyes are going or they are kind of making little jokes about … it being a date… (JD16, female)

It's [surgery] so very male dominated. It's probably the one thing with medicine that is still very male dominated. And a girl can sometimes not be taken very seriously." (JD14, female)

Another participant suggested that women may feel less confident compared with their male counterparts:

…I think as a woman you are more likely to be cautious and potentially undersell your skills or not be as assertive (JD05, female)

A few participants had experienced racism from patients; this participant recalled a distressing experience during foundation training when he was racially abused by the father of a patient after raising concerns about suspicious bruising following a non-accidental injury. The father subsequently made a complaint. During this incident, the study participant indicated that he felt poorly supported by his senior who was more concerned about the complaint than the racism:

So he [the father] called me a dirty 'P', etc, said he was going to murder me, he was going to hunt me

down, etc, etc, etc… then he said I want to see a white doctor… (JD01, male)

A few female participants experienced the intersection of racism and sexism, as this participant highlights:

I had a consultant who I worked under and he used to make comments. He said both of these at one point when I was under his supervision; 'I'm not racist but…' and then he would say something that was a very generalising, stereotyping comment. He said, 'I'm not sexist but…' (JD17, female)

### Blaming and shaming

Over half the study participants reported having worked in a culture or team where it was commonplace to blame and shame individuals, often junior staff, for any medical errors. Participants highlighted that such a culture of finger pointing, with no ownership of collective responsibility or addressing of systemic failings, detracted from opportunities for team learning and highlighted the importance of creating safe and supportive environments:

That's very prevalent in the NHS and it happens a lot. People use shame and guilt a lot. Seniors use that all the time…overtly undermining you in that subtle way that is just enough to know that you're being undermined….If there is a cock-up or there is a problem, the consultant, the boss or the professor should say, 'This has happened.' This could be done in a group type setting. 'A patient has complained about this. I suggest we do this, that and the other to try and fix it.' This should be done without naming names, shaming or any of that. I think dealing with it as a team is always more helpful rather than name and shame individuals. (JD03, male)

The blame culture and associated lack of support was most distressing when participants were blamed and poorly supported to manage critical incidents, as the following participant recalls:

I think that has a massive impact on your ability to cope with adverse events then because in that unit, if anything went wrong, you were really concerned that the midwives would just try and pin it on you and say you were the only one involved and it wasn't a system problem or a team problem but it was an individual. (JD13, Female)

The following participant, a foundation year 2 doctor recalls feeling isolated, blamed, abandoned and poorly supported by the team and consultant after a serious critical incident at work. The event was psychologically traumatic and had long-term implications for her career, resulting in regular instances of outward distress at work, and later triggering an episode of depression:

There were all these other things and ultimately, you just think, 'I feel quite alone, vulnerable and I'm an F2.' That was when I really started… I can feel myself

getting teary. It's six or seven years later and it's still painful. You then kind of question everything you then do because I just didn't trust the system. I didn't trust my seniors to have my back… You just put a brave face on and think that your own mental struggle is no different to anyone else's, so why should you deserve to be in tears in a corridor? Oh, I was in linen cupboards crying on most shifts. It was awful. (JD08, female)

Some participants suggested that the blaming and shaming culture was sometimes employed as a defence against organisational or systemic failings:

…she just stood in the middle of the ward and berated me and told me I was awful and again…if she was ever on-call we wouldn't bother her with anything because anything that goes wrong, it's just your fault and no-one else's fault and she couldn't recognise that it was actually a systemic issue. (JD01, male)

The following participant indicated that she was considering leaving the NHS to emigrate and work abroad due to the growing demand and expectations on doctors, and the culture of blame:

I don't know whether I will stay in England because of the state of the NHS and because of the growing demand, expectation and blame culture on doctors. (JD17, female).

### Lack of support

Over half the study participants had experienced poor or no psychological or emotional support while working as a junior doctor. They described a lack of empathy and support, particularly from other team members (eg, nurses and midwives), consultants and supervisors. This was particularly in response to managing the emotional fallout from distressing events, bullying in the workplace, or having to taking time off work for physical or mental health problems and bereavement. Feeling unsupported left them isolated and more vulnerable, often compounding work-related stress and distress.

### Lack of support: critical incidents

Participants reflected on the importance of feeling supported, especially when managing the psychological impact of critical incidents such as complaints, or the death of a patient:

When bad things happen, if you don't get the support from colleagues, you're not in an environment where people can share … then it's just infinitely more difficult. (JD15, female)

I've had a recent death on the gynae ward which I found really difficult, partly because I felt…I suppose a bit isolated in dealing with it. (JD05, female)

## Lack of support: supervisors and deaneries

Some participants felt that they were not always able to discuss the stresses and strains of the job with their supervisors and were reluctant to disclose or discuss their mental health issues:

> So my current clinical supervisor is lovely and he's very approachable. We have a good rapport but I wouldn't have gone to him, I think, with the issues I've had. (JD17, female)

The qualities of a supervisor or consultant are important and being approachable and empathetic are key to feeling supported and crucial for patient safety, for clinical input and reassurance, as this participant highlights:

> The people who are supervising makes a massive difference in terms of whether they're supportive or if they still like to humiliate and make you feel completely incompetent… If you feel that your senior is unapproachable or just gives you a really hard time, it stresses you out and you don't want to approach them which, I suppose, isn't good for patient safety either. (JD22, female)

The following participant who took time off work for chronic stress found that her deanery was unsupportive of her need to take time out:

> I mean I've cried so much and I feel disgusted and betrayed and heartbroken by my own deanery, you know, [yeah] that I've been with all these years, with how they've treated me, I feel absolutely worthless. (JD20, female)

## Lack of support: discontinuity in teams

When junior doctors are on 4-month rotations, early on their training, building relationships and accessing support from colleagues can be challenging. They can often be geographically distanced from their support networks—friends, family and partners, which may be particularly isolating when struggling at work with high workloads and understaffing, as the following participant illustrates:

> One of the things that I did want to say to you about the enormous stress that is caused by having to pick up and move to the other side of the country, sometimes when you haven't even chosen to go there because it's like a national application system (yeah) you rank where you want to go but you are never guaranteed a placement close to your family and friends (mmm) so often so many junior doctors and especially F1s and 2s feel so isolated (yeah) you know far away from their families and friends, starting this really difficult new job where often they feel under-prepared and short-staffed and you know so many people struggle. (JD18, female)

The following participant highlighted that the lack of continuity in multidisciplinary teams can be challenging and that the absence of belonging can add to a sense of isolation:

> It feels very individualistic because teams are no longer a thing in the NHS with the shift work pattern. You come in at 8 o'clock and you leave at 5 o'clock … but every day I'm with a different team and this means there is no sense of belonging necessarily. (JD03, female)

## Stigma and culture of appearing invulnerable

The majority of participants described a culture of needing to appear invulnerable within medicine, where work-related stress and distress were normalised and taking time off for physical or mental health is poorly tolerated. Participants felt there was considerable stigma associated with mental illness, work-related stress and burnout, which delayed help seeking for some, and often led to many participants working when they were unwell, a concept termed 'presenteeism', related to guilt and fear of employment-related discrimination, negative social judgement, or being admonished or shamed, as the following participants illustrate:

> …the kind of culture of you're not allowed to be ill, you're not allowed to suffer, then there's a stigma attached and you end up going to work. (JD09, female)

> I think there's definitely still a stigma about mental health in medicine, definitely, and actually particularly stress-related problems because I think so much … kind of gets directed back towards the individual, like why are you so stressed and maybe you're not resilient enough… (JD16, female)

Some participants reported feeling ashamed to admit that they were unwell and internalised this as a personal failure with their perceived inability to cope with the demands and emotional impact of the job:

> To show that you're struggling is a sign of failure or a weakness and that's really hard to be open about. (JD08, female)

> It's far easier to tell people that you know you've got arthritis or you've got cancer than you've got depression and I think to talk about any mental health things, there is still that stigma attached in medicine…. (JD19, female)

> There is very much the attitude of just grinning and bearing it; so, whatever you're feeling is somehow invalid or invalidated because you just have to grin and bear it and just carry on. (JD17, female)

A few participants reflected on the culture of intolerance towards vulnerability in which junior doctors are acculturated to work while unwell (presenteeism) and consequently feel ashamed to disclose or admit to any physical or mental health condition:

> I think that there's a sense of shame about leaving your colleagues fighting fires, when you're off unwell.

I definitely come in when I've been more unwell than I maybe should. (JD07, female)

## DISCUSSION

Our findings report the varied sources of emotional and psychological distress which derive from stressful working cultures and conditions which are often interconnected and cumulative. Toxic work cultures, including being bullied, blamed, shamed, inadequately supported and feeling isolated were commonly reported sources of distress, with some junior doctors experiencing sexism and racism from patients and other staff. This study found that being bullied or blamed and shamed, coupled with a lack of support, were often the tipping point for junior doctors, leading to depression and persistent suicidal thoughts. Such experiences were also reported as a key 'push' factor in intention to leave the profession or work abroad.

Distress arising from the responsibility of managing high clinical workloads was compounded by the toxic work factors listed previously, together with an intolerance of vulnerability and feeling poorly supported, particularly in response to critical incidents, which often left the junior doctors feeling emotionally isolated. This is notable, as feeling isolated is a risk factor for suicide.[24] Participants' experiences of bullying are evidenced by a recent survey which found that 10% of doctors in training reported bullying and harassment from managers, and 20% reporting bullying and harassment from colleagues.[25] Evidence also suggests that experiencing or witnessing bullying among NHS staff has been found to be detrimental to psychological health and job satisfaction and is also associated with intention to leave the profession.[26] Burnout is principally attributed to prolonged exposure to chronic emotional and interpersonal stressors[27]; working in toxic cultures is therefore likely to contribute towards burnout.

These findings reinforce the urgent need for significant culture change and improvements to working conditions and team working in the NHS. A GMC report identifying factors impacting on the mental health and well-being of medical students and doctors stated that a culture change through leadership is necessary to promote positive mental health in NHS doctors and other staff.[28] One approach suggested for achieving cultural and leadership change is a shift to more meaningful, inclusive and impactful leadership and management styles which take account of diverse work cultures.[29]

Leaders need to promote support systems and enact processes which enable staff to disclose their vulnerability;[28] for example, through effective debriefs, and deployment of workplace interventions such as the use of Schwartz rounds which provide permission and a safe space for healthcare staff to talk about the range of feelings and challenges which arise during their work while also fostering connectedness to others.[30] NHS organisations and medical schools may need to rethink the current over-reliance on resilience training,[10] which focuses the gaze on individuals and how well they can adapt and cope with adversity during their training, yet ignores the systemic causes of distress, and often leaves medical students and doctors feeling ashamed of their perceived inability to do this. Participants in this study often felt blamed for systemic or organisational failings which were beyond their control.

Organisations have responsibility for establishing supportive and compassionate work environments, which is often contingent on tackling the root causes of toxic work cultures. Such cultures are often symptomatic of systemic problems and the imbalance between demand and resources available (ie, staff shortages and underfunding). Consultants and other leaders also require support from their organisations in order to fulfil the demands and expectations of their role and by ensuring values are aligned and embedded across the workplace.

According to data collected in 2019, more women are gaining places on medical undergraduate courses (59% female) and attaining training posts (54% female); such figures are not reflected in the 32% of women who progress to consultancy,[31–33] representing a significant drop-off. While career breaks to have children provide one explanation, the experiences of female participants in this study may suggest other factors at play, and echo research investigating barriers to progression in medicine for women.[31–33] This research suggests a potential link between male dominated or hegemonic workplaces, toxic work cultures and a culture of invulnerability, but further longitudinal, quantitative evidence will be required to establish more definitive relationships between gender and work culture. It can also be noted that male participants in this study also reported bullying, being shamed and humiliated by male and female consultants and therefore interventions to improve cultures and leadership are required irrespective of gender.

Evidence has found that, in addition to bullying and sexism experienced by female participants, workplace cultures were found to be the most significant barrier to progression and job satisfaction—the communication of subtle negative cues and signals to women (microaggressions) prohibits promotion and career progression, and contributes to the lack of confidence and self-esteem experienced by many women.[34] Discrimination towards women in medicine is also evidenced by lower pay.[35]

The internalisation of such messaging might explain the manifestation of imposter syndrome experienced by some female participants. Although imposter syndrome is experienced by women and men, evidence suggests women are more likely to experience this at different stages of their career.[36 37] Crucially, social support, validation of success and positive affirmation have been found to be protective against imposter syndrome, which is noteworthy since imposter syndrome has been identified as a risk factors for burnout and suicide.[38] Consequently, there needs to be an emphasis on recruiting, promoting

and supporting a diversity of women to senior leadership positions.[39]

Stigma persists within medicine, as illustrated by the experiences of study participants. Shame or guilt induced presenteeism is commonplace in doctors and is often enmeshed within the professional identity;[40 41] 42% of doctors in England and 47% in Wales reported having recently attended work despite not feeling well enough to perform their duties.[42] This is detrimental to the well-being of individuals, linked to the development of depression and jeopardises patient safety.[43–45] Supervisors and leaders need to ensure that there are supportive and confidential mechanisms and practices facilitating taking sufficient time off when required, so that doctors avoid feeling guilty, judged or ashamed; workforce shortages and rota gaps need to be addressed as they compound the pressure to attend work while sick. Evidence has shown that stigma associated with appearing distressed, vulnerable or mentally unwell are significant barriers to help seeking, particularly among healthcare professionals;[46] any delays to help seeking for anxiety, depressive, bipolar-related disorders or psychosis are associated with worse mental health outcomes,[46] reinforcing the need for proactive interventions to challenge stigma and its impact in the workplace, particularly for a population of healthcare staff who are vulnerable to distress.

The study participants highlighted that the absence of support from managers and teams was distressing and isolating. Working in effective and supportive teams requires staff to have shared objectives, values and continuity, necessitating planning and changes to rotas which can help to ensure junior doctors feel part of a team.[28] There is substantial evidence that working in teams which lack continuity in staff or shared objectives is detrimental to the mental health and well-being of staff and impacts negatively on patient care and safety.[47 48] Junior doctors need to feel valued, capitalising on existing support infrastructure and assets (ie, teams, supervisors and colleagues) to provide clinical and psychological support (eg, interprofessional debriefs for processing and validating feelings). Organisations will need to provide the training and coaching to develop effective and cohesive teams while also meeting recommendations set out in existing guidance to ensure the clinical, learning, training and emotional needs of doctors are met.[49]

There are many potential solutions to the problem of work-related stress: some lie in existing evidence for effective interventions and policies, others are discussed in this paper. Additionally, the National Institute for Health and Care Excellence in England suggests that organisations which invest in and promote the mental health and well-being of their staff result in increased job satisfaction, staff retention, improved productivity and performance and reduced staff absenteeism.[50] This paper highlights the potential benefits to staff if such guidance were actioned while also potentially conferring benefits to patients.[51]

Although whistleblowing is protected by law[52] and endorsed in NHS policies,[53] some study participants feared and had experienced reprisal from whistleblowing and experienced bullying and harassment from senior staff. More needs to be done to support and enable staff to whistle-blow safely where appropriate, and to tackle the attitudes and culture which directly or indirectly sanction divisive and discriminatory practices.

There is substantial evidence linking longer working hours with mental ill health and reduced patient safety; junior doctors who worked over 55 hours a week were more than twice as likely to report a common mental health problem[54] and increased likelihood of making a medical error.[55] This highlights the importance of providing suitable facilities for sleep, rest and social spaces, and facilitating social interaction and access to collegial support. The BMA's Fatigue and Facilities charter sets out UK-wide minimum standards for basic facilities in healthcare organisations to ensure all doctors have access to places and time to rest and sleep and access to nutritious food and drink; these essential requirements for their work need to be implemented universally.[56]

This study has some limitations. There is a notable disparity in gender, with a higher proportion of female doctors taking part. We received higher expressions of interest from female (n=37) participants compared with male (n=22), with a higher proportion of female participants then agreeing to participate in an interview. The interest in this study and increased willingness to come forward and talk about their experiences, among female participants, may reflect evidence indicating that female doctors are more likely to experience distress, with increased rates of suicide evidenced in young female doctors. The higher proportion of female participants may also reflect gendered help-seeking behaviour for mental ill health, evidenced in the wider population.[57] Another potential limitation relates to the lower number of participants who reported having had suicidal thoughts (n=5) and participants who had made self-harm attempts (n=2).

## CONCLUSION

Notably, participants reflected on the culture of intolerance towards vulnerability in which junior doctors are acculturated to work while unwell (presenteeism) and which poses a key barrier to seeking help or support. Experiences of bullying, discrimination and a blame and shame culture highlight the need for a culture shift within medicine to offer more supportive and compassionate work environments. There needs to be greater recognition of the components and cumulative effects of toxic workplaces and stressors related to high workloads, compounded by staff shortages and a lack of access to clinical and emotional support. Organisations need to capitalise on existing potential support from teams, supervisors and colleagues to ensure staff are supported and to reduce isolation in the workplace. The opportunity to provide feedback about supervisors could facilitate the provision of high-quality support for all junior

## Open access

doctors. More needs to be done to support and enable staff to 'whistle-blow' where indicated, and to tackle the attitudes and culture which directly or indirectly sanction divisive and discriminatory practices. Practical changes to physical working conditions also need to be addressed, with provision of suitable spaces to eat, sleep, rest, debrief and connect with colleagues.

**Author affiliations**
¹Institute of Applied Health Research, University of Birmingham College of Medical and Dental Sciences, Birmingham, UK
²Research Department of Primary Care and Population Health, University College London, London, UK
³School of Social Policy, University of Birmingham, Birmingham, UK
⁴Department of Organizational Psychology, Birkbeck University of London, London, UK
⁵Oxford University Hospitals NHS Trust, Oxford, UK
⁶Faculty of Medicine and Health, University of Leeds, Leeds, UK
⁷London School of Hygiene & Tropical Medicine, London, UK
⁸Institute of Applied Health Research, University of Birmingham, Birmingham, UK
⁹Psychiatry and Behavioral Sciences, University of Manchester, Manchester, UK
¹⁰School of Medicine, Keele University, Keele, UK

**Acknowledgements** The authors thank all junior doctor participants who kindly gave their time to share their experiences of working in the NHS. We also wish to thank members of the patient and public involvement and engagement and public involvement and engagement group who provided valuable input throughout the study.

**Contributors** RR, CC-G, MB, KT, AG, AKT, MVH, LA and JM: substantial contributions to conception and design. RR, FK, CC-G, MB, KT, AG, AKT, LA and MVH: acquisition of data, or analysis and interpretation of data; drafting the article or revising it critically for important intellectual content; and final approval of the version to be published.

**Funding** The study was funded by NIHR Research for Patient Benefit (PB-PG-0418–20023).

**Competing interests** None declared.

**Patient consent for publication** Not required.

**Ethics approval** Ethical approval was granted by the University of Birmingham and Health Research Authority (reference number 19/HRA/6579).

**Provenance and peer review** Not commissioned; externally peer reviewed.

**Data availability statement** All data relevant to the study are included in the article. This study has not received ethical approval to share confidential data with any third party other than the study research team.

**ORCID iDs**
Ruth Riley http://orcid.org/0000-0001-8774-5344
Kevin Teoh http://orcid.org/0000-0002-6490-8208

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
