## [Reviewer comments · BMJ Open]

ARTICLE DETAILS

TITLE (PROVISIONAL)	The sources of work related psychological distress experienced by United Kingdom-wide foundation and junior doctors: a qualitative study
AUTHORS	Riley, Ruth; Buszewicz, Marta; Kokab, Farina; Teoh, Kevin; Gopfert, Anya; Taylor, Anna; Van Hove, Maria; Martin, James; Appleby, Louis; Chew-Graham, Carolyn

VERSION 1 – REVIEW

REVIEWER	Miharu Nakanishi Research Center for Social Science & Medicine, Tokyo Metropolitan Institute of Medical Science, Japan
REVIEW RETURNED	19-Sep-2020

GENERAL COMMENTS	The manuscript investigated source of work-related psychological distress among medical doctors who may be at an early stage of their careers. The findings can be notable for international audience as well as health policy decision makers in England, pointing out the toxic organisational cultures in healthcare. The reviewer found that all factors presented in this study are shared with healthcare organisations in my country, where some medical schools had manipulated exam scores to favour male candidates. Some additional discussions may be helpful to increase the implications from the findings for the cultural change toward more supportive and compassionate work environments. Abstract, participants Page 5 line 17-20: description may be inserted on how many females participated INTRODUCTION Page 7 line 20-23: As global population ages and healthcare needs continue growing, implications from the findings can contribute to sustainable healthcare workforce worldwide. DISCUSSION Page 15 line 42-47: male-dominated culture may correlate with intolerance of vulnerability, that requires young doctors adapt oneself to the toxic cultures, high workload and poor working conditions. Page 16 line 20-30: It would be again highlighted here that 'participants often felt blamed for failings which they attributed to organisational and systemic problems'. The toxic cultures may help healthcare organisations ignore their needs to establish supportive and compassionate work environments.
---

	Page 16 line 31-40: It would be noted that consultants and other healthcare staff who 'survived' by adapting themselves to current work environments also have needs for support for the cultural change.
--	---

REVIEWER	Dr Sally Pezaro Coventry University, United Kingdom
-----------------	--

REVIEW RETURNED	02-Oct-2020
-------------

GENERAL COMMENTS	Dear colleagues This is a very well written article on an interesting study which would be a great contribution to the literature in this regard. I hope the following points serve to strengthen this work further prior to publication.  -In the abstract it would be useful to refer to the fact that there were four main themes, and list what these were rather than present a list of findings on their own. -At the end of the introduction, it would be useful to outline a more direct and specific objective and/or research question, rather than simply say what this article reports on -In methods, it would be useful if you could expand some more on your epistemology/ontological perspective here. I am excited that this work is qualitative and would like to see more context given in this regard. - Would be useful to expand upon the reflexivity of the research team in line with COREQ - Findings are interesting and well presented. - In the discussion, your arguments in relation to women do not go far enough in securing the point you are trying to make. At the end of these paragraphs I am left thinking...So what?.. -At times you refer to this as a report - is it a report or an article or a paper? Need to be consistent. - Please expand on suggesting future research directions Thank you for doing this important work.
--

VERSION 1 – AUTHOR RESPONSE

Reviewer: 1
Reviewer Name: Miharu Nakanishi

Reviewer: 2
Reviewer Name: Dr Sally Pezaro

Reviewer: 1
Institution and Country: Research Center for Social Science & Medicine, Tokyo Metropolitan Institute of Medical Science, Japan

Reviewer: 2

Institution and Country: Coventry University, United Kingdom

Reviewer: 1

Comments to the Author

The manuscript investigated source of work-related psychological distress among medical doctors who may be at an early stage of their careers. The findings can be notable for international audience as well as health policy decision makers in England, pointing out the toxic organisational cultures in healthcare.

We are appreciative and grateful for this feedback.

The reviewer found that all factors presented in this study are shared with healthcare organisations in my country, where some medical schools had manipulated exam scores to favour male candidates. Some additional discussions may be helpful to increase the implications from the findings for the cultural change toward more supportive and compassionate work environments. We thank the reviewer for this comment and we have now positioned our research within an international context, particularly in relation to hegemonic toxic work cultures (p. 5).

Abstract, participants

Page 5 line 17-20: description may be inserted on how many females participated

This has now been added

INTRODUCTION

Page 7 line 20-23: As global population ages and healthcare needs continue growing, implications from the findings can contribute to sustainable healthcare workforce worldwide.

Thank you for this comment. We have now added a statement highlighting the sustainability factors impacting on doctors' (and other healthcare staff) mental health. We include references from other countries to reflect that such contexts and pressures are evidenced and are experienced universally (p. 5).

DISCUSSION

We thank the reviewer for these pertinent comments and we have now incorporated these points within the discussion (p.14) .

Page 15 line 42-47: male-dominated culture may be associated with intolerance of vulnerability; this requires young doctors to adapt to toxic cultures, high workload and poor working conditions.

Page 16 line 20-30: It would be again highlighted here that 'participants often felt blamed for failings which they attributed to organisational and systemic problems'. The toxic cultures may help healthcare organisations ignore their needs to establish supportive and compassionate work environments.

Page 16 line 31-40: It would be noted that consultants and other healthcare staff who 'survived' by adapting themselves to current work environments also have needs for support for the cultural change.

Reviewer: 2

Comments to the Author

Dear colleagues

This is a very well written article on an interesting study which would be a great contribution to the literature in this regard. I hope the following points serve to strengthen this work further prior to publication.

We thank the reviewer for this comment.

-In the abstract it would be useful to refer to the fact that there were four main themes, and list what these were rather than present a list of findings on their own.

This has now been done (p.4).

-At the end of the introduction, it would be useful to outline a more direct and specific objective and/or research question, rather than simply say what this article reports on

A specific objective has now been added (p.6).

-In methods, it would be useful if you could expand some more on your epistemology/ontological perspective here. I am excited that this work is qualitative and would like to see more context given in this regard.

Thank you for this comment. We have now added a section on reflexivity which provide insight into the epistemological/ontological perspectives taken by the two key researchers (p.7-8).

- Would be useful to expand upon the reflexivity of the research team in line with COREQ

We have now included a separate reflexivity section in the qualitative tradition (p.7-8).

- Findings are interesting and well presented.

Thank you.

- In the discussion, your arguments in relation to women do not go far enough in securing the point you are trying to make. At the end of these paragraphs I am left thinking...So what?..

We thank you for this comment.

In this paper we refer to a number of solutions aimed at promoting more women into leadership positions and importance of valuing and supporting effective leadership styles. We have now referenced NICE guidance to promote mental health in the workplace and which urgently needs to be actioned in the NHS. (p14-15?)

-At times you refer to this as a report - is it a report or an article or a paper? Need to be consistent.

We have now used the term 'paper' throughout.

- Please expand on suggesting future research directions

Based on the findings reported in this study and its paired paper 'protective factors reported among junior doctors', the study team suggest that policy changes and the implementation of guidance (such as NICE) and following through on West & Coia's (2019) recommendations are required to reduce occupational distress experienced by junior doctors.

Thank you for doing this important work.

Thank you for appreciating the importance of this study.

Reviewer: 1

Competing interests 1: None declared

Reviewer: 2

Competing interests 1: None declared

VERSION 2 – REVIEW

REVIEWER	Miharu Nakanishi Tokyo Metropolitan Institute of Medical Science, Japan
REVIEW RETURNED	21-Jan-2021

GENERAL COMMENTS	Overall the revisions are adequately made in response to our previous comments. A short suggestion may be helpful for policy decision makers on how to support consultants and other leaders (page 14 line 40-43), i.e. to reduce their conflict with established values and attitudes that are inconsistent with the organisational cultural changes.
--

REVIEWER	Dr Sally Pezaro Coventry University, United Kingdom
REVIEW RETURNED	18-Jan-2021

GENERAL COMMENTS	Thank you for conducting this important work.
---